The Phanerozoic diversification of silica-cycling testate amoebae and its possible links to changes in terrestrial ecosystems

Lahr Daniel J.G. 1 dlahr@ib.usp.br
Bosak Tanja 2
Lara Enrique 3
Mitchell Edward A.D. 3 4
1 Department of Zoology, Institute of Biosciences, University of São Paulo , Rua do Matão, São Paulo , Brazil
2 Department of Earth, Atmospheric and Planetary Sciences, Massachussetts Institute of Technology , Cambridge, MA , USA
3 Laboratory of Soil Biology, University of Neuchatel , Neuchatel , Switzerland
4 Jardin Botanique de Neuchâtel , Neuchatel , Switzerland
De Baets Kenneth
Electronic publication date: 2015 Sep 8
Publication date: 2015
Volume: 3
Electronic Location ID: e1234
Received 2015 May 29; Accepted 2015 Aug 19
Copyright: © 2015 Lahr et al.
Copyright year: 2015
Copyright holder: Lahr et al.
License: This is an open access article distributed under the terms of the Creative Commons Attribution License, which permits unrestricted use, distribution, reproduction and adaptation in any medium and for any purpose provided that it is properly attributed. For attribution, the original author(s), title, publication source (PeerJ) and either DOI or URL of the article must be cited.
License URL: https://creativecommons.org/licenses/by/4.0/

Keywords: Amoebozoa, Evolution of eukaryotes, Conquer of terrestrial environment, Molecular clocks, Testate amoebae, Silica cycle, Coupled carbon and silica cycles, Vase shaped microfossils, Ancestral state reconstruction

Funding: Wikimedia.ch fund FAPESP Young Investigator Award 2013/04585-3 NASA Astrobiology Institute award NNA13AA90A NASA Exobiology award 6927184 CNPq-MIT seed fund #551952/2011-3 University of Neuchatel The digitalization of Eugene Pénard’s slide collection was made possible by a Wikimedia.ch fund. DJGL is funded by a FAPESP Young Investigator Award (2013/04585-3); TB is funded by NASA Astrobiology Institute award NNA13AA90A and NASA Exobiology award 6927184. Traveling was partly funded by a CNPq-MIT seed fund (#551952/2011-3) to DJGL and TB. Sampling and microscopic analyses were funded by a “fonds de donation” from the University of Neuchatel through a post-doctoral fellowship to DJGL in collaboration with EADM and EL. The funders had no role in study design, data collection and analysis, decision to publish, or preparation of the manuscript.

==============================
The terrestrial cycling of Si is thought to have a large influence on the terrestrial and marine primary production, as well as the coupled biogeochemical cycles of Si and C. Biomineralization of silica is widespread among terrestrial eukaryotes such as plants, soil diatoms, freshwater sponges, silicifying flagellates and testate amoebae. Two major groups of testate (shelled) amoebae, arcellinids and euglyphids, produce their own silica particles to construct shells. The two are unrelated phylogenetically and acquired biomineralizing capabilities independently. Hyalosphenids, a group within arcellinids, are predators of euglyphids. We demonstrate that hyalosphenids can construct shells using silica scales mineralized by the euglyphids. Parsimony analyses of the current hyalosphenid phylogeny indicate that the ability to “steal” euglyphid scales is most likely ancestral in hyalosphenids, implying that euglyphids should be older than hyalosphenids. However, exactly when euglyphids arose is uncertain. Current fossil record contains unambiguous euglyphid fossils that are as old as 50 million years, but older fossils are scarce and difficult to interpret. Poor taxon sampling of euglyphids has also prevented the development of molecular clocks. Here, we present a novel molecular clock reconstruction for arcellinids and consider the uncertainties due to various previously used calibration points. The new molecular clock puts the origin of hyalosphenids in the early Carboniferous (∼370 mya). Notably, this estimate coincides with the widespread colonization of land by Si-accumulating plants, suggesting possible links between the evolution of Arcellinid testate amoebae and the expansion of terrestrial habitats rich in organic matter and bioavailable Si.

Introduction

Si is a major rock-forming element with a cycle that influences the growth of primary producers and carbon burial in the oceans (Sarmiento, 2013). Over geological time scales, the biogeochemical cycles of carbon and silica are linked through the weathering of continents, which dissolves Si from rocks and delivers it to the oceans (Wilkinson & Mitchell, 2010). On shorter time scales, Si cycles vigorously in soils and forms a soil pool that is 2–3 orders of magnitude larger than the Si pool in living terrestrial biomass (Cornelis et al., 2011). Thus, plant-microbe-mineral interactions that control this sizeable pool of soil Si ultimately control the availability of dissolved Si and the delivery of Si to the oceans (Conley, 2002). Plants are thought to be the major contributors to the terrestrial cycling of Si because they can promote the weathering of rocks, accumulate Si from the soil solution and biomineralize amorphous Si in the form of phytoliths (Alexandre et al., 1997; Cornelis et al., 2011). Phytoliths released from dead plant matter can form a pool with a lower turnover rate relative to other forms of biogenic silica (Alexandre et al., 1997). This pool comprises more than 90% of biogenic Si delivered to rivers (Cary et al., 2005) and is the main source of reactive Si in soils. Research in the past three decades has revealed much about the role of plant-derived biogenic Si in the terrestrial cycling of Si (Conley, 2002). In contrast, the contribution and the long term history of Si-biomineralizing microbial groups in terrestrial ecosystems are less well understood (Wilkinson & Mitchell, 2010). Many microbial eukaryotes use silica to build external and internal skeletons, and have molecular mechanisms for Si uptake. Up to 77 genes regulated by silicic acid in the diatom Phaeodactylum tricornutum have orthologs in the genomes of other eukaryotes, including Opisthokonts, Viridiplantae and other “Chromalveolates” (Sapriel et al., 2009); see also Fig. 1. The genes implicated in silica metabolism may have been exchanged among eukaryotic clades through lateral gene transfer, as demonstrated for choanoflagellates and diatoms (Marron et al., 2013). In terrestrial systems, testate amoebae (i.e., amoebae that construct shells) are among the most abundant and conspicuous organisms that use silica. The existing studies show that: (1) testate amoebae can contribute up to 10% of biogenic silica in some tropical soils and rivers (Cary et al., 2005); and (2) the annual incorporation of Si by testate amoebae can in some cases match the amounts of Si released by plant phytoliths (Aoki, Hoshino & Matsubara, 2007; Wilkinson, 2008; Sommer et al., 2012; Puppe et al., 2014). These observations, as well as the long evolutionary history of testate amoebae (Lahr, Grant & Katz, 2013), suggest a role for testate amoebae in the terrestrial silica cycle and motivate this study.

Figure 1 A simplified tree of eukaryotes indicating that biomineralization is a convergent feature.

The main supergroups are indicated by the different colors and the lineages in bold contain biomineralizers. Backbone of tree is based on relationships in Katz (2012), dotted lines represent uncertainty.

There are several groups of unrelated testate amoebae. The two most prevalent and abundant in terrestrial environments are the euglyphid and the arcellinid testate amoebae. Both inhabit the same environments—bodies of fresh water, soils, peatlands and other humid microhabitats—and have approximately the same sizes, with the majority of species being between 30–300 µm long or wide. However, the two groups are vastly divergent genetically and historically. Euglyphids include about 800 species and are in the super group Rhizaria (Fig. 1). These organisms produce thin, pointed, non-anastomosed pseudopods and almost all extant lineages in the group are silica biomineralizers. Thus, biomineralization is likely ancestral in the group. Owing to the preservation of siliceous shells, euglyphids have a reliable fossil record that goes back at least 30–50 million years (Foissner & Schiller, 2001; Barber, Siver & Karis, 2013). The arcellinid testate amoebae encompass about 2,000 species and are in the super group Amoebozoa (Fig. 1). These amoebae produce rounded, blunt pseudopods and have a great diversity of shell compositions—organic, agglutinated and biomineralized. The fossil record of arcellinids is much older than that of euglyphids, and there is consensus that some vase shaped microfossils dating back to the Neoproterozoic (ca. 750 mya) belong to the arcellinids (Porter & Knoll, 2000; Bosak et al., 2011; Lahr, Grant & Katz, 2013; Strauss et al., 2014).

Biomineralization of silica in testate amoebae occurs in many different ways. The shell is always constructed shortly before cell division: a new shell is produced through the aperture of the older shell. After cell division, one daughter cell stays in the old shell and the other daughter cell inherits the new shell (Hedley & Ogden, 1974). Most euglyphids produce silica scales in the cytoplasm, presumably taking up dissolved Si and depositing it as amorphous silica via silica deposition vesicles (Hedley & Ogden, 1974; Anderson, 1994; Gröger, Lutz & Brunner, 2008). The scales, which are typically shorter than 10 µm and thinner than 2 µm, are then used as building blocks to construct the shell. The specific literature refers to these types of building blocks produced by testate amoebae as idiosomes. A small number of arcellinids use a similar strategy—Lesquereusia, Netzelia, and especially Quadrulella (Fig. 2A) are three genera known to produce silica idiosomes (Anderson, 1987; Anderson, 1989; Anderson, 1994; Meisterfeld, 2002). Netzelia is able to precipitate idiosomes, but is also known to deposit silica around ingested particles, including starch and various minerals, and then use these particles to build the daughter shell (Anderson, 1987; Anderson, 1989). Quadrulella, on the other hand, produces its shell entirely of square siliceous idiosomes. Many arcellinids use siliceous particles and mineral grains scavenged from the environment as unmodified building blocks named xenosomes (Difflugia (Fig. 2B) and Heleopera are well-known examples (Meisterfeld, 2002; Châtelet, Noiriel & Delaine, 2013)). Others are able to lightly modify siliceous particles either by dissolution or deposition (e.g., Nebela (Fig. 2C) and related genera (Padaungiella, Argynnia (Fig. 2D)), as well as the insertae sedis Lesquereusia (Anderson, 1987; Anderson, 1989).

Figure 2 Examples of arcellinids shell composition.

(A) Quadrulella subcarinata Gautier-Lièvre, 1957 constructs the shell using square particles of amorphous Si that are endogenously produced from dissolved silica (idiosomes). Specimen from Sphagnum collected in Welgevonden Game Park, Limpopo province, South Africa. (B) Difflugia acuminata Ehrenberg builds its shell from agglutinated diverse particles, in this case, the organism used both centric (white arrow) and pennate (black arrow) diatom shells, along with other smaller particles. (C) Nebela marginata Penard uses a mixture of particles with some additional biological silica deposition, such as scales scavenged from euglyphids (oval and circular plates as the one indicated by the white arrow), and pennate diatoms (black arrow). (D) Argynnia dentistoma, this specimen has used a mixture of flat environmental mineral particles and rounded euglyphid scales to construct the shell. (B–D): Specimens from Eugene Penard’s collection, deposited at the Natural Museum of Geneva; photos taken by Thierry Arnet–Wikimedia document. Scale bars 30 µm.

Both classical and modern studies report the usage of euglyphid scales by arcellinid amoebae of the Hyalospheniidae family (Leidy, 1879; Penard, 1902; Deflandre, 1936; Douglas & Smol, 2001; Meisterfeld, 2002). These amoebae reportedly obtain silica plates by preying on euglyphids, and then use the stolen scales to build the shell (Deflandre, 1936)—a phenomenon we name kleptosquamy (Fig. 3). Here, we record several stages of this phenomenon in Padaungiella lageniformis that preys upon Euglypha sp. Next, we ask whether kleptosquamy is ancestral in the hyalosphenid testate amoebae and use this to determine the order in which hyalosphenids and euglyphids emerged. To better time the rise of biomineralization in hyalosphenids, we also provide a novel molecular clock reconstruction of the arcellinids. Finally, we discuss the implications of the revised molecular clock in light of broader evolutionary and biogeochemical trends.

Figure 3 An example of kleptosquamy in the arcellinid Apodera vas (larger shell), obtained from predation upon the euglyphid Sphenoderia valdiviana (two smaller individuals).

The two species occur together in Sphagnum magellanicum mosses around Laguna Esmeralda, in Argentinian Tierra del Fuego. The larger scales (arrows) in the test of A. vas can clearly be matched to the ones produced by S. valdiviana.

Material and Methods

Microscopical observations

Samples of Sphagnum sp. were collected in Les Pontins peat bog in Canton Bern, Switzerland (47°7′39.11″N; 6°59′27.35″E). Microscopic observations were made using an Utermöhl chamber (Cat #435025; HydroBios, Kiel, Germany) on an Olympus IX81 inverted microscope equipped with oil immersion Differential Interference Contrast optics (20×–40×–60×–100×). All images were recorded by an Olympus DP-71 camera.

Ancestral state reconstructions

We have performed ancestral state reconstructions on the topologies from molecular reconstructions of two recently published phylogenies (Kosakyan et al., 2012; Oliverio et al., 2014). Each reconstruction is based on a distinct set of molecular data (Cox1 and SSU rDNA respectively). Ancestral state reconstruction was performed in the program Mesquite (Maddison & Maddison, 2007) using parsimony as an optimality criterion, for the single character kleptosquamy, with possible states present, absent or unknown.

Molecular clock reconstructions

Molecular clock reconstructions (MCR) were performed using PhyloBayes 3.3 (Lartillot, Lepage & Blanquart, 2009). We used the final tree and alignment for SSU rDNA small subunit ribosomal gene published by Lahr, Grant & Katz (2013) as a tree onto which we calculated divergence times. Calibration points were the 6 opisthokont fossils also used by Parfrey et al. (2011), whereas the Arcellinida calibration point is based on the fossil Paleoarcella athanata (type specimen HUPC #62988), described in Porter, Meisterfeld & Knoll (2003). The dating of sedimentary rock for this fossil was an ash bed 2 m above the fossils, calculated by U–Pb zircon chronology (Karlstrom et al., 2000) (Table 1). The opisthokont fossils used by Parfrey et al. (2011) are congruent with those proposed and justified for animals by Benton et al. (2015). With additional data present in the current tree, it was possible to use the Chuar group fossils as a calibration point for the actual last common ancestor of arcellinids (Porter & Knoll, 2000), rather than the divergence between arcellinids and other naked amoebae, as in Parfrey and colleagues (2011). One alternative run was also generated incorporating the three additional Meso- and Cenozoic fossils as calibration points within the Arcellinida, as suggested by Fiz-Palacios, Leander & Heger (2014): origin of the Centropyxis genus (termed “node B” in Fiz-Palacios, Leander & Heger (2014)) was set to the split between Hyalosphenia papilio and Arcella hemisphaerica, with lower and upper bounds at 736–220 mya, origin of hyalosphenids (“node C”) was set to the split between Padaungiella lageniformis and Hyalosphenia elegans with soft bounds at 736–100 mya; origin of genus Arcella (“node D”), calibrated the clade containing A. hemisphaerica and A. vulgaris WP with soft bounds at 105–100 mya. We did not include the fourth calibration point suggested by Fiz-Palacios and colleagues (Lesquereusia–Difflugia divergence) because the Lesquereusia SSU rDNA is not available. Fortunately, Fiz-Palacios and colleagues (2014) have tested their dataset for sensitivity to this particular calibration point and have determined that its inclusion does not significantly modify the final result. We have performed MCRs by running two independent chains with a burn-in factor of 100 until the effective size of samples was above 50 and the maximum discrepancy between chains was below 0.3. These parameters are suggested as values for an “acceptable run” by the PhyloBayes manual. We used soft constraints on the calibration dates to account for uncertainty in the fossil dates as advocated by several researchers (e.g., Donoghue & Benton, 2007; Parfrey et al., 2011). The use of soft constraints requires a model of birth–death for the prior on divergence times. We performed reconstructions using both the GTR and the CAT-GTR models for nucleotide substitutions. We have performed MCR using three distinct models for rate distributions: two auto correlated models (CIR and lognormal) as well as the uncorrelated gamma multipliers model. We performed comparisons for model fit by computing Bayes Factors by thermodynamic integration under the normal approximation, as discussed in Phylobayes 3.3 manual, and recommended in Lartillot & Philippe (2006). In order to do so, a variance–covariance matrix was obtained using the program estbranches (part of multdivtime package (Thorne & Kishino, 2003)), with input parameters calculated in baseml (part of the PAML package (Yang, 2007)), following instructions by Rutschmann (2005).

Table 1 Summary of calibration points used of molecular clock reconstructions.

Dates are in millions of years.

Clade	Fossil	Taxa used for delimitation	Max date	Min date	
Amniota	Westlothiana	Gallus gallus and Homo sapiens	400	328.3	
Ascomycetes	Paleopyrenomycetes	S.s pombe and P. chrysosporium	1,000	400	
Endopterygota	Mecoptera	A. mellifera and D. melanogaster	350	284.4	
Animals	sponge biomarkers	O. carmella and C. capitata	3,000	632	
Bilateria	Kimberella	B. floridae and C. capitata	630	555	
Vertebrates	Haikouichthys	B. floridae and H. sapiens	555	520	
Arcellinida	Paleoarcella	A. hemisphaerica and H. sphagni	3,000	736	

Results

Microscopical observations

An individual Padaungiella lageniformis was isolated while preying upon a specimen of Euglypha sp. The Euglypha cytoplasm had been almost completely ingested at the stage of isolation (Figs. 4A–4C). We observed the P. lageniformis removing and ingesting siliceous plates from the prey organism’s shell for around 10 min (Fig. 4D). Immediately afterwards, the individual deposited siliceous plates in the inner part of the “neck” of its shell, parallel to the aperture (Fig. 4E) and moved large cytoplasmic debris in the same direction (Figs. 4E and 4F). The organism was constructing a plug in the aperture, which became visible at the end of this activity that lasted for approximately one hour. There was no indication that any of the plates were dissolved in the cytoplasm: all previously ingested plates were kept intact and moved towards the aperture. After one hour, the amoeba added multiple additional layers to the previously laid down barrier (Figs. 4G–4I). Though the scales formed most of the barrier, other types of debris were added as well. After about two hours of actively plugging the aperture, the amoeba encysted, presumably with digestive function (Figs. 4J and 4K). The amoeba remained encysted for at least two more hours. Resumed observations approximately 12 h later revealed that the amoeba had emerged from the digestive cyst and re-ingested all siliceous plates (Fig. 4L). However, the organism had discarded the yellowish-brown types of debris (Fig. 4L). The siliceous plates did not appear to be separated from any other cytoplasmic structures by membranes. Many gathered around the nucleus at times. Two days after the initial ingestion, the amoeba went into a second resting cyst (it is relevant to note that the amoeba was maintained in an environmental sample and had access to ample food items), where it remained for over 24 h, but did not exhibit other relevant changes. We discontinued observations approximately 80 h after initial observation (4 days).

Ancestrality of kleptosquamy

Observations of kleptosquamy and the associated behavioral attitudes enable evolutionary interpretations in other closely related hyalosphenid testate amoebae. The conspicuous plug made of scales created by Padaungiella, as well as the presence of modified euglyphid scales in the shell is observed in most other hyalosphenids, including the genera: Apodera, Certesella (Meisterfeld, 2002), Porosia (Fig. 5A), and finally Nebela (Fig. 5B). One other genus has shells that contain small, possibly siliceous scales with undetermined origin: Physochila and Argynnia (Fig. 2D, (Vucetich, 1974)) were shown not to be closely related to hyalosphenids (Gomaa et al., 2012). Considering the most current molecular data available, Padaungiella is a basal lineage (Lara et al., 2008; Heger et al., 2011; Gomaa et al., 2012; Kosakyan et al., 2012; Lahr, Grant & Katz, 2013). Other four genera (Apodera, Certesella, Porosia, Nebela) are able to re-use scales obtained from euglyphids and three others (Quadrulella, Hyalosphenia and Alocodera) do not. We have performed a parsimony based ancestral reconstruction of character states in both topologies available (based on mitochondrial and nuclear genes, Fig. 6). Under any of the two scenarios scenario, kleptosquamy appears in the ancestral hyalosphenid, and is lost twice: once in the genus Quadrulella, which biomineralizes its own silica scales and once in Hyalosphenia, which builds entirely proteinaceous scales without mineral parts. In the scenario of Oliverio et al. (2014), kleptosquamy is lost 3 times because the genus Hyalosphenia is not monophyletic. The non-monophyly of Hyalosphenia has no effect on the ancestral character state for hyalosphenids as a whole (Fig. 6).

Figure 4 Kleptosquamy in Padaungiella lageniformis.

(A) Lateral view of P. lageniformis ingesting cytoplasm of Euglypha sp., beginning of observations (T = 0). (B) View closer to the bottom of the plate, where the teardrop shaped apertural scales of the Euglypha individual are visible (white arrow), and other already ingested plates are in P. lageniformis cytoplasm (black arrow, T = 22 min). (C) A distinct optical section from B, showing a region in the Euglypha shell where the roughly hexagonal body plates (white arrow) were removed by the P. lageniformis, note that here the apertural scales are not present on this side (T = 22 min). (D) Accumulation of plates from Euglypha in the cytoplasmic region of P. lageniformis close to the aperture (black arrows), in the cytoplasm, plates are easily visible when in profile view (T = 22 min). (E) Early stage of apertural plug construction, the P. lageniformis has laid down two scales (black arrow) in a parallel orientation to the aperture (T = 23 min). (F) The organism begins to add other debris to the plug (black arrow, t = 24 min). (G) Debris particles had been added to the plug, notice vesicles of yellowish-brown material in the cytoplasm (black arrows), these are later added to the plug (T = 1 h 13 min). (H) Yellowish-brown debris moves closer to the aperture (black arrows, T = 1 h 13 min). (I) All debris particles finally added to the apertural plug (black arrows, T = 1 h 32 min). (J) After the apertural plug is finished, the cell goes into a cyst (T = 1 h 52 min). (K) Whole view of digestive cyst (T = 1 h 52 min). (L) Emergence of cyst 12 hours later, many scales are visible in the cytoplasm (black arrows), they were recollected from the plug. Other types of particles were discarded. Scale bar = 20 µm (B–J) and 50 µm (A, K, L).

Figure 5 Evidence of kleptosquamy in other hyalosphenid genera.

(A) A specimen of Porosia bigibbosa in a digestive cyst, with an apertural plug constructed partly with siliceous scales (white arrows). Specimen from mosses collected on an erratic boulder near the Merdasson river, Neuchâtel, Switzerland. (B) A specimen of Nebela marginata, about to enter the digestive cyst, presenting also an apertural plug constructed with a layer of siliceous scales (white arrows), among others. Specimen from Sphagnum collected in Les Pontins bog, Canton Bern, Switzerland. Scale bar =50 µm.

Molecular clock reconstructions

To determine the origin of hyalosphenids, we generated a dated tree for the arcellinids (Fig. 7). In order to do so, we used the previously established opisthokont calibration points and a conservative calibration point for the minimum date of origin of the Arcellinida—this calibration point is used conservatively as calibrating the entire Arcellinida, as opposed to the less inclusive family Arcellidae as suggested by affinities in the original description (Porter, Meisterfeld & Knoll, 2003). In this reconstruction, we used the uncorrelated gamma multipliers model for the distribution of divergence times. This is because in our model fit analyses, this model yielded the largest Bayes Factor (logBF interval of 18.7–26.9, against 12.8–14.5 for CIR model and 16.6–24.7 for lognormal model). The Bayes Factor is one of many proposed methods to measure the appropriateness of a given model for the data at hand, and a larger BF indicates a better model fit (Lartillot & Philippe, 2006). Hence, all results discussed are based on that model. The reconstruction ran for circa 55,000 cycles until convergence between the two chains was achieved.

Figure 6 Ancestral state reconstruction of kleptosquamy in the hyalosphenid genera.

The backbone of each cladogram is one of two most current hyalosphenid phylogenies, based on distinct sets of genes. Colors along the tree branches represent how states changed through evolution for this character.

Figure 7 Comparison between dated phylogenies of Arcellinida based on molecular clock reconstructions using distinct sets of calibration points.

Both reconstructions were based on a 109 taxon, 914 positions alignment. Taxa that are not relevant for the present discussion have been collapsed for clarity. The reconstruction on the left uses a single arcellinid calibration point (indicated), and other 5 calibration points inside the Opisthokonta. The reconstruction on the right uses the previous 6 calibration points plus 3 additional arcellinid calibration points. Although the mean value for node times can be quite different, both reconstructions are within the 95% confidence interval of each other (indicated by shaded horizontal bars).

Our reconstruction stands in sharp contrast with another recent molecular clock reconstruction of the arcellinids (Fiz-Palacios, Leander & Heger, 2014). The two most likely reasons for this are: (1) although Fiz-Palacios and colleagues used a part of the same dataset used here, we focused on the SSU rDNA partition and not on the protein coding partition; (2) we included mostly opisthokont fossils as calibration points, but Fiz-Palacios and colleagues used a number of Meso- and Cenozoic microfossils as calibration points for internal families of arcellinids. To test the influence of these hypotheses, we implemented the calibration points suggested by Fiz-Palacios in our framework. This yielded an additional tree (Fig. 7, right). This tree is very similar to the tree obtained by Fiz-Palacios and colleagues, with all origins of groups tending to appear at younger dates. For instance, the origin of arcellinids as a whole shifts from 944 mya to 600 mya using the Meso- and Cenozoic fossils. This is representative of a general trend throughout the tree. Therefore, the distinct dates obtained in the reconstruction presented in Fig. 7 probably do not stem from focusing on the SSU rDNA partition, but rather from the use of distinct calibration points. Hence, the interpretation of fossils is paramount in defining which result is more likely to reconstruct the actual history of Arcellinida.

Discussion

Kleptosquamy can be inferred as an ancestral character state in hyalosphenids (Fig. 6), i.e., the last common ancestor of all extant hyalosphenids was able to re-use euglyphid scales. Hyalosphenid biology is not well understood, because most attempts to culture these organisms have failed. Strains that have been maintained for a certain time (Nebela collaris) had to be fed with fast-growing species of euglyphids, such as Euglypha hyalina (Meisterfeld, pers. comm., 2013). For instance, one cannot say with certainty whether a hyalosphenid is able to construct the shell without any euglyphid scales. This caveat undermines the interpretation of klepstosquamy as ancestral in the group, for this reason, we clearly establish that our working hypothesis is that scaled euglyphids appeared before hyalosphenids.

The fossil record of euglyphids is quite sparse and does not currently allow accurate timing of their evolution. The very well documented microfossils of Scutiglypha from diatomaceous earth demonstrate that modern genera have existed for at least 15 million years (Foissner & Schiller, 2001). More recently, Eocene microfossils have unambiguously pushed the fossil record of euglyphids back to 50 million years ago (Barber, Siver & Karis, 2013), including members of the genus Scutiglypha (Euglyphidae). Older records of shells are much more difficult to interpret, as the conditions of shell preservation make the separation between arcellinids and euglyphids ever more difficult: because of intense convergence, pseudopods would be the only reliable way of separating arcellinids and euglyphids, but these are usually not preserved in the fossil record (Bosak et al., 2011; Lahr et al., 2014). For instance, some vase shaped microfossils described from the Chuar group, especially Melicerion poikilon, Bonniea spp. and Bombycion micron have morphological characteristics that are compatible with euglyphid testate amoebae: the typical vase shape, thin walls, terminal aperture, homogeneously shaped and sized scales and an apparent siliceous composition. However, an arcellinid origin cannot be excluded, Quadrulella, a modern arcellinid, shares all those features (Porter & Knoll, 2000; Porter, Meisterfeld & Knoll, 2003). Hence, new discoveries of exceptionally preserved and properly described arcellinid and euglyphid fossils are necessary to inform interpretations of origin and diversification (Bosak et al., 2011; Dalton et al., 2013; Fiz-Palacios, Leander & Heger, 2014; Strauss et al., 2014).

The dated reconstruction presented here uses a single-locus and external calibration points to the arcellinids and places the origin of hyalosphenids in the Paleozoic (about 370 mya, with a 95% confidence interval that extends from the Neoproterozoic to Triassic). This is in marked contrast to the recent reconstruction of arcellinid history by Fiz-Palacios, Leander & Heger (2014), which placed the origin of hyalosphenids in the Cretaceous, about 130 mya (with a 95% confidence interval between the Devonian and the Eocene). These authors used a very similar dataset, but included both the SSU rDNA and five additional protein coding genes (both analyses are based on the dataset published by Lahr, Grant & Katz, 2013). The 240 million year difference between the two reconstructions is significant and may lead to very distinct implications. The use of the same calibration points as Fiz-Palacios and colleagues, combined with our search strategy, produced a tree that is very similar to the results of Fiz-Palacios et al. and estimate a Cretaceous rise of hyalospheniids. The additional calibration points used are controversial, some are based on fossils whose descriptions have not clearly established syngenicity with the matrix and may be contaminants (Farooqui et al., 2010; Kumar, 2011); others come from amber and the identity of organisms is established using optical microscopy alone (Schmidt, Schönborn & Schäfer, 2004; Schmidt et al., 2006; Girard et al., 2011). The inclusion of these fossils as calibration points led to an interesting scenario interpreted by Fiz-Palacios, Leander & Heger (2014): that hyalosphenids are an ancient lineage that diversified when the complex peatland environments became available. The caveat is that many aspects of the identities of fossils used as calibration points remain to be clarified—this does not mean that the interpretations are incorrect.

The new molecular clock reconstruction (Fig. 7) suggests that various testate amoebae including hyalospheniids, the aquatic Arcella + Netzelia clade, as well as the soil dwelling Trigonopyxis + Bullinularia clade have diversified after the mid-Devonian. In contrast to the preceding periods, when plant cover was restricted to moist habitats, the Late Devonian and the Carboniferous saw the diversification of plants that were well adapted to life on land, with deeper roots and soil forming capabilities (e.g., Gibling & Davies, 2012; Kenrick et al., 2012). These plants formed extensive forests, established their own, humid environments, and produced abundant organic matter as well as soils (Kenrick et al., 2012), matching the appearance of the Bullinularia clade. These evolutionary events likely influenced the Si cycling on land as well due to two main factors: (1) The root systems of the Late Devonian/Carboniferous plants are thought to have promoted silicate weathering (e.g., Algeo, Scheckler & Maynard, 2001), and (2) tree-like Lycopodiophyta, Equisetales and liverworts, plants whose modern relatives can accumulate as much or more Si than grasses (Hodson et al., 2005), were abundant in forest ecosystems. Our new molecular clock reconstruction and the coinciding sequence of evolutionary and ecological changes that have been documented in the fossil record inspire questions. Did the release of Si from plants and the accumulation of Si in the plates of biomineralizing testate amoebae as well as in various predatory species strengthen the links between the C and the Si cycles on land? The annual rate of biosilification by testate amoebae was shown to be of the same order as the uptake rate by trees (Aoki, Hoshino & Matsubara, 2007; Sommer et al., 2012; Puppe et al., 2014) and the size of the Si pool in testate amoebae increases with vegetation development in some early ecosystem successions (Puppe et al., 2014).

To move forward from here, we need to (i) better understand Si metabolism across protist groups, including deeper understanding of physiological aspects; (ii) obtain reliable fossil evidence for the appearance and diversification of Si-metabolizing lineages; (iii) improve the molecular clock by expanding the molecular database, using appropriate numerical models and carefully checking the reliability of the fossil record, (iv) improve constraints on the contribution of Si-precipitating organisms to the cycling of Si and C in terrestrial systems. The combination of these efforts is challenging, but can be met with a combination of approaches including molecular phylogeny, biogeochemistry, and paleontology.

Supplemental Information

Supplemental Information 1 Raw data used for analyses

Alignments, trees and command lines used in molecular clock reconstructions.

Click here for additional data file.

We thank Piet-Louis Grundling for collecting the Sphagnum sample from which the Quadrulella illustrated in Fig. 2A was found. We also thank the associate editor as well as two reviewers for relevant criticism of the manuscript.

Additional Information and Declarations

Competing Interests

Author Contributions

Data Availability

The authors declare there are no competing interests.

Daniel J.G. Lahr conceived and designed the experiments, performed the experiments, analyzed the data, contributed reagents/materials/analysis tools, wrote the paper, prepared figures and/or tables, reviewed drafts of the paper.

Tanja Bosak analyzed the data, contributed reagents/materials/analysis tools, wrote the paper, reviewed drafts of the paper.

Enrique Lara analyzed the data, wrote the paper, reviewed drafts of the paper.

Edward A.D. Mitchell conceived and designed the experiments, analyzed the data, contributed reagents/materials/analysis tools, wrote the paper, reviewed drafts of the paper.

The following information was supplied regarding data availability:

The alignments and trees used, as well as a document with instructions and command lines used for running analyses, are available as a supplemental file.

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
