# Peer review of "The Phanerozoic diversification of silica-cycling testate amoebae and its possible links to changes in terrestrial ecosystems"

_PeerJ, doi:10.7717/peerj.1234_

## Round 0.1 · original submission · Minor Revisions

The submitted manuscript contains interesting observations and results, which are well-worth publishing. However, several points need to be addressed to make the study easier to follow and more scientifically reproducible. The following points need to be particularly addressed in the revision:

Molecular clock methodology

• Justification and implementation of calibrations: As it is now well-established your calibrations should be justified more adequately for better reproducibility and more accurate results (see also comments by reviewer 1). You rely here on calibrations of previous studies (Fiz-Palacios et al. 2014; Parfrey et al. 2011), which does fully justify their calibrations following these requirements. Furthermore, in the meantime phylogenetic interpretation of fossils might have changed and dating or correlation with the latest geological time-scale of individual fossils might have improved. There are now better constraints available for some of the nodes that you have using. This is well illustrated for some Metazoan nodes (Metazoa, Vertebrata, Amniota), which you used in your study by Benton et al. (2015). Furthermore, other calibrations might also have been superseded in the meantime. Part of your study also demonstrated that molecular clock estimates can be particularly sensitive to the calibrations used, making such a priori screening and justification of calibrations even more important (Warnock et al. 2015). You also allude to this in the text that some of the points used by Fiz-Palacios et al. (2014) are , but do not describe the problems with them in greater detail. It is not entirely clear how you (or the previous studies) selected your maximum constraints as these need to equally justified. Proper justification is crucial if you want to perform scientific hypothesis testing rather than storytelling. You could use the new justified constraints for various animals nodes (Benton et al. 2015), the same should be done for remaining calibrated nodes. These justifications could be added the main text or should at least be added to the supplementary material. Calibrations are considered well-justified when the following steps are fulfilled (see Parham et al. 2012):
o Museum numbers of specimen(s) that demonstrate all the relevant characters and provenance data should be listed. Referrals of additional specimens to the focal taxon should be justified.
o An apomorphy-based diagnosis of the specimen(s) or an explicit, up-to-date, phylogenetic analysis that includes the specimen(s) should be referenced.
o Explicit statements on the reconciliation of morphological and molecular data sets should be given.
o The locality and stratigraphic level (to the best of current knowledge) from which the calibrating fossil(s) was/were collected should be specified.
o Reference to a published radioisotopic age and/or numeric timescale and details of numeric age selection should be given.

• Sensitivity analyses with calibrations: I was a bit surprised that only two analyses were performed. It is by now well established that adding calibrations to additional nodes or different implementation of calibrations can impact on divergence time estimates(Clarke et al. 2011; Warnock et al. 2015; Warnock et al. 2012). Particularly, the root is very crucial for molecular dating and needs to be properly justified(Warnock et al. 2012). For this reason, it is necessary to test the influence of such different implementation of calibrations onto your dataset more adequately. Such a type of sensitivity analysis was done to test the terrestrialization of arthropods(Rota-Stabelli et al. 2013). Additional sensitivity analyses are called for if you want to make interpretations and conclusions of broader relevance.
• Confidence intervals: the confidence intervals are sometimes very broad, which makes it difficult to clearly demonstrate a link with changes in terrestrial ecosystems. However, such a link could make sense from other types of evidence, making it possible to erect a hypothesis which could be further test with new discoveries. Personally, I think it is better to aim for an accurate time-scale which does not yet have the precision that we want, than having a precise time-scale which lacks the accuracy that we need.
Hypothesis of Fiz-Palacios and colleagues: you need to discuss in more detail the “interesting scenario” presented by Fiz-Palacios et al. (2014) and why in your opinion the identity of these fossils is uncertain (see comments by reviewer 1)

Kleptosquamy: Reviewer 2 was a bit skeptical of the widespread occurrence and significance of kleptosquamy as you only observed it only under the microscope in some forms. It would therefore be called for to explain the significance in greater detail (can single observations be extrapolated to entire species or genera), possibly expressing yourself more carefully on its implications or explaining how this should be further corroborated by additional studies.
Title: Considering that the link with changes in terrestrial ecosystems is rather interpretative; it might be better to drop this aspect from the title and focus on observations you made (e.g., kleptosquamy) or things you did.

Comparisons with plant evolution: you mainly compare the divergence of your amoebae with studies on the fossil record, which are now slightly outdated. Additional studies have been done of the evolution of plants and their link with terrestrial ecosystems(Gibling & Davies 2012; Kenrick et al. 2012; Retallack 1997; Retallack & Huang 2011). These have focused particularly on carbon cycling, but the influence of root depth and soil formation on weathering should also impact the Si cycle. Furthermore, more recent studies on the diversification of land plants have been done based on molecular clock methodologies(Clarke et al. 2011) and the fossil record(Silvestro et al. 2015). More up-to-date studies should be discussed in this context.

Comparison with other protozoa with silica tests: Considering that diatoms are the most closely related group independently evolving silica tests in terrestrial ecosystems, it would be interesting to compare the results obtained here with the diversification of diatoms (compare reviewer 2).

In addition to the comments by the reviewers, the following points also need to be addressed in the revision:
Line 44-45: please add reference(s) for the following statement “Si is major rock-forming element … and carbon burial in the oceans.”
Line 57-58: please add reference(s) for the following statement “Research in the past three decades … in the terrestrial cycle of Si.”
Line 81-82: I guess you mean “almost all extant lineages” instead of “almost all lineages” as you can´t possibly how this was distributed in the geological past
Line 107: I suggest to add “well-known” before “examples”
Line 142: Please properly format “Porter, Meisterfeld & Knoll, 2003” (not capitalized)
Line 161: Parfrey et al. (2011) were not the first or last to advocate the use of soft-constraints(Benton et al. 2009; Donoghue & Benton 2007). It is well-known that the choice and implementation fossil calibrations can have severe effects on divergence time estimates.
Line 221: You need to be more specific and explain in greater detail what you mean with conservative calibration.
Line 242: “the use of distinct calibration points”: yes, this is well-known, but it does not only depend on the used calibrations points, but also how they are implemented. It would therefore make sense to compare different implementations of your calibration point. Furthermore, this also highlights the need to justify the maximum and minimum age for each calibration point in more detail.
Line 263: “pseudopods would be the only reliable way”: I guess it would be worth telling that they are usually not preserved in the fossil record
Line 270: “better described arcellinid fossil record”: this is a bit confusing as it sounds now as you mean that the already discovered specimens have been poorly described, but I guess what you mean to say is that “new discoveries of exceptionally preserved and properly described fossils are necessary” to disentangle this
Line 291-292: yes, the fossil used should be more justified (e.g, Parham et al. 2012), but this also need to be done for the points used in your study.
Line 298: “Selden & Edwards, 1989; Gray 1993”: I guess new discoveries might have been made after this time (see comments above)
Used references:
Benton MJ, Donoghue PC, Asher RJ, Friedman M, Near TJ, and Vinther J. 2015. Constraints on the timescale of animal evolutionary history. Palaeontologia Electronica 18:1-106.
Benton MJ, Donoghue PCJ, and Asher RJ. 2009. Calibrating and constraining molecular clocks. In: Hedges BS, and Kumar S, eds. the Timetree of Life. Oxford: Oxford University Press, 35-86.
Clarke JT, Warnock RCM, and Donoghue PCJ. 2011. Establishing a time-scale for plant evolution. New Phytologist 192:266-301.
Donoghue PCJ, and Benton MJ. 2007. Rocks and clocks: calibrating the Tree of Life using fossils and molecules. Trends in Ecology & Evolution 22:424-431.
Fiz-Palacios O, Leander BS, and Heger TJ. 2014. Old Lineages in a New Ecosystem: Diversification of Arcellinid Amoebae (Amoebozoa) and Peatland Mosses. PLoS ONE 9:e95238.
Gibling MR, and Davies NS. 2012. Palaeozoic landscapes shaped by plant evolution. Nature Geosci 5:99-105.
Kenrick P, Wellman CH, Schneider H, and Edgecombe GD. 2012. A timeline for terrestrialization: consequences for the carbon cycle in the Palaeozoic. Philosophical Transactions of the Royal Society B 367:519-536.
Parfrey LW, Lahr DJG, Knoll AH, and Katz LA. 2011. Estimating the timing of early eukaryotic diversification with multigene molecular clocks. Proceedings of the National Academy of Sciences 108:13624-13629.
Retallack GJ. 1997. Early Forest Soils and Their Role in Devonian Global Change. Science 276:583-585.
Retallack GJ, and Huang C. 2011. Ecology and evolution of Devonian trees in New York, USA. Palaeogeography, Palaeoclimatology, Palaeoecology 299:110-128.
Rota-Stabelli O, Daley Allison C, and Pisani D. 2013. Molecular Timetrees Reveal a Cambrian Colonization of Land and a New Scenario for Ecdysozoan Evolution. Current Biology 23:392-398.
Silvestro D, Cascales-Miñana B, Bacon CD, and Antonelli A. 2015. Revisiting the origin and diversification of vascular plants through a comprehensive Bayesian analysis of the fossil record. New Phytologist 207:425-436.
Warnock RCM, Parham JF, Joyce WG, Lyson TR, and Donoghue PCJ. 2015. Calibration uncertainty in molecular dating analyses: there is no substitute for the prior evaluation of time priors. Proceedings of the Royal Society B: Biological Sciences 282.
Warnock RCM, Yang Z, and Donoghue PCJ. 2012. Exploring uncertainty in the calibration of the molecular clock. Biology Letters 8:156-159.

·

Basic reporting

The formatting of the reference list is highly inconsistent. Some references are even incomplete, e.g. lines 437(!), 441, 449. Journal names in lower case vs. in capitals (lines 338, 368). I wonder if there is an initial manuscript check in PeerJ; this should have returned to authors before sending for Review.

Experimental design

No comments

Validity of the findings

No comments

Additional comments

This is a very well-written and nicely presented study. I enjoyed reading it and I hope to see it published soon. I suggest, however, a minor revision of the manuscript.

lines 140-150 : It should be precisely explained which fossils were used for calibration, including the precise age of the fossils, including up-to-date references of geologic age for all localities.

line 215-216: "non-monophyly" might be replaced by "polyphyly" or "paraphyly", depending on what is appropriate.

lines 289-291: This vague circumscription should be specified. Please mention which fossils Fiz-Palacios et al. used and what the "interesting scenario" actually is. And why in the author's opinion the identity of the fossils used by these authors is uncertain.

Fig. 7 and Fig. 7 caption should definitely show which fossils were used for analyses.

Reviewer 2 ·

Basic reporting

The manuscript is really well written - consisting nice figures and high-quality pictures. The structure of the text and quality of English is very good.

Experimental design

The data presented are really interesting, however the error bars of the calibration are very wide. I am aware this is a kind of standard in case of such fragmentary paleontological information. This is evident that there are not a lot of fossil records of the testate older than Quaternary that not well recognised, therefore determination of time of diversification is extremely difficult. I appreciate the effort of authors to estimate molecular clock using so scarce information - as so far only paleontological record can help. This uncertainty should be appropriately addressed in the Discussion.

Validity of the findings

The data are highly interesting and important for the better understanding of Protist evolution, with the focus Si-accumulation. Authors are exploring the issue of preying of Padauginella langeniformis on Euhypha sp. plates. Despite it is a really nice observation, it was done rather not in the systematic way, and it purely based on microscopic observation, not supported by the physiological analysis. My question is how this single observation can be extrapolated to so general statements on kleptosquamy and then molecular clock calibration ? Despite I am really enthusiastic on the results obtained, this single-approach looks methodologically controversial.

Additional comments

The manuscript provides a very intriguing idea on the possible origin of the idiosomic, silica-cycling testate amoebae and kleptosquamy. Furthermore, authors tried to calibrate the origin of Arcellinida. They related the origin of silica-cycling testate amoebae to the existence of the Late Devonian/Carboniferous Si-accumulating plants. Actually, I am curious how the testate amoebae silica-cycling divergence time is related to the diatoms appearance? It is not widely discussed in the text. One of the main conclusions is related to Si-accumulating plants colonisation, but diatoms are rather omitted in this context.

Some details are hard to see in the pictures presented. Especially in Figure 5 plates are hard to observe in the plug despite white arrows. Also, for the figure 6 - how presence of kleptosquamy was identified for all these species? Is this subjectively guessed or related to the literature?

---

## Round 0.2 · Minor Revisions

Thanks you for taking into account our suggestions and recommendations. The manuscript is as good as accepted, but I found some minor typos and formatting issues, which need to be addressed before publication (as they are hard to change once the manuscript is officially accepted). I append an annotated PDF. I apologize for the inconvenience.

I would also like to clarify my concerns with molecular clock studies (which I also use for some of my research), which is not related with me being not a microbiologist, but a more general concern which should be taken into account in molecular clock studies in all fields.I totally agree that the placement and choice of your calibration fossils is generally accepted, although such fossils are accepted as long as somebody might interpret them anew. I am well aware that morphological phylogenetic analyses for fossils are hard to come by in microbiology and it is similar for many invertebrate groups I work on, but it does not mean one should not strive to mention what the main morphological characters are used to place them in this group (as you do now in your manuscript). I was more alluding to other steps (e.g., age dating of fossils) and I should have been more clear. For example Parfrey et al. (2011) states to have used the geological time-scale of 2009, but in the meantime newer more-up-to-date geological timescale have been published, which could potentially have shifted the age constraints for certain stages or ages up to a couple of million years and could impact the molecular clock estimates. Although I doubt this will impact your molecular clock study much considering the large confident intervals on your divergence estimates. Anyway, resubmission deadlines at PeerJ are just approximate, so if you had needed more time to revise your manuscript then that is no problem (usually this is requested via a short message to the editors). I just wanted to mention these concerns (also for future studies), which you are apparently mostly aware aware off, but did not explicitly mention in the manuscript before. Considering the scope of your article, it might be better in your specific case to use the constraints used in the previous studies to make them more directly comparable, but this needed to more explicitly stated as it is done now. Thank you for your understanding.

---

## Round 0.3 · accepted · Accept

Thank you for making these final changes.